# Built Environment Accessibility and Disability as Predictors of Well-Being among Older Adults: A Norwegian Cross-Sectional Study

**DOI:** 10.3390/ijerph20105898

**Published:** 2023-05-20

**Authors:** Grace Katharine Forster, Leif Edvard Aarø, Maria Nordheim Alme, Thomas Hansen, Thomas Sevenius Nilsen, Øystein Vedaa

**Affiliations:** 1Department of Neuromedicine and Movement Science, Norwegian University of Science and Technology, NTNU, NO-7047 Trondheim, Norway; 2Department of Health Promotion, Norwegian Institute of Public Health, NIPH, NO-5015 Bergen, Norway; leifedvard.aaro@fhi.no (L.E.A.); oystein.vedaa@fhi.no (Ø.V.); 3Department of Health and Functioning, Western Norway University of Applied Sciences, HVL, NO-5063 Bergen, Norway; maria.nordheim.alme@hvl.no; 4Department of Mental Health and Suicide, Norwegian Institute of Public Health, NIPH, NO-0456 Oslo, Norway; thomas.hansen@fhi.no (T.H.); thomassevenius.nilsen@fhi.no (T.S.N.); 5Centre for Welfare and Labour Research, Oslo Metropolitan University, NO-0170 Oslo, Norway; 6Promenta Research Center, University of Oslo, NO-0317 Oslo, Norway; 7Department of Psychosocial Science, University of Bergen, UiB, NO-5015 Bergen, Norway; 8Voss District Psychiatric Hospital NKS Bjørkeli, NO-5705 Voss, Norway

**Keywords:** healthy ageing, older adults, well-being, disability, built environment, quality of life

## Abstract

Knowledge about the influence environmental factors have on well-being is important to deliver policies supporting healthy ageing and sustainable health equity. An under-researched question is whether and how the built environment plays a role on well-being among older adults with disabilities. This study explores the relationship between built environment accessibility and disability on psychosocial well-being among older adults. Data were used from the Norwegian Counties Public Health Survey collected during February 2021 in Møre and Romsdal county (*N* = 8274; age = 60–97, mean = 68.6). General linear modelling was performed to examine the relationship and interaction between built environment accessibility (services, transportation, and nature) and disability on psychosocial well-being (quality of life, thriving, loneliness, and psychological distress). Higher levels of disability and poorer accessibility were each significantly related to lower psychosocial well-being across all variables (*p* < 0.001). Significant interaction effects were observed between disability and built environment accessibility on thriving (*F*(8, 5936) = 4.97, *p* < 0.001, η^2^ = 0.006) and psychological distress (*F*(8, 5957) = 3.09, *p* = 0.002, η^2^ = 0.004). No significant interaction effects were found for quality of life and loneliness. These findings indicate good built environment accessibility is associated with thriving and reduces psychological distress among older adults with disabilities. This study supports and extends previous findings on the importance of accessible and equipped environments for well-being and may aid policy makers when planning built environments to foster healthy ageing among this population group.

## 1. Introduction

Governments worldwide have been called on to deliver policies ensuring sustainable health equity and good health and well-being for people with disabilities [1]. Differences in health outcomes among the general population and people with disabilities are profound, with the latter more likely to have poorer health and higher risk of mortality and experience more social injustice [2,3]. Older adults with disabilities particularly experience more problems in maintaining independence and functioning than younger populations [1]. Ageing is a risk factor and approximately 34% of people aged ≥60 years live with some form of disability, compared to 16% across all ages [1]. Incidence among older adults is predicted to rise due to population ageing and increasing prevalence of non-communicable diseases. Health inequities experienced by people with disabilities are commonly due to contextual factors arising from environmental stressors that create unfair living conditions and exacerbate health disparities [2].

Scholars have long raised concern over the magnitude of the health gradient in high-income nations such as Norway and attribute this to environmental factors, such as where and how people live, rather than monetary wealth [4]. Non-communicable diseases account for ten of the major reasons for disability among older adults [5,6]. At the same time, interventions have reduced overall mortality from the main causes of age-related disease, and life expectancy has increased by an average of four years over the past three decades [7]. However, longevity for many is occurring parallel to more years lived with disabilities [3]. Disability and healthy ageing are each defined by the World Health Organization (WHO) as participation outcomes from the interaction between a person’s health and functioning, together with personal and environment factors. Healthy ageing describes the ability to maintain meaningful participation to enable well-being in older age, whilst disability describes reduced participation compared to others on an equal basis [3,8]. Physiological decline is a natural part of ageing, where the majority of people will experience reductions at some point in life [8,9]. Healthy ageing with a disability depends on whether environmental and personal factors foster participation, despite the potential onset of health problems [3,8].

Disability often affects daily life beyond the impairment itself, and secondary health conditions are common. Evidence indicates significant associations between disability and indicators to psychological and social (psychosocial) well-being, such as low quality of life [10], psychological distress [11], poor self-rated health [10], reduced happiness/thriving [12], loneliness [10,13], depression, and suicidal behavior [14]. The built environment describes human-made structures, features, and facilities physically part of the lived environment [15]. On the basis of healthy ageing, pathways between poorer psychosocial well-being and disability may be connected with a lack of sufficient resources in the environment to support varying levels of functioning, which in turn can negatively impact health behaviors and participation and increase social gradients in health [3,8]. Research has documented that people with disabilities experience difficulties accessing fitness facilities [16], using public transportation [17], and navigating neighborhood green spaces and walking paths [18]. Whilst more evidence is needed specifically on older adults with disabilities [19], previous research suggests that certain characteristics within built environments are important for health and participation. For example, accessible services, transport, and nature are linked to attending health appointments, maintaining daily independence [17,20,21,22], mobility and physical activity [20,22,23,24], social connectiveness [22,24,25,26], reduced risk of mortality [27], and improved psychosocial well-being [28,29,30].

Improved health behavior at a population level from changes to the built environment may deliver greater public health benefits than clinical health interventions alone [31]. In their model of “age-friendly environments” for healthy ageing and in the context of the Decade of Health Ageing [32], the WHO identify eight key domains to prioritize in public health promotion and planning to support a life course approach to ageing and disability. Three of the domains concerning nature, transport, and services refer to characteristics in the built environment shown to support healthy ageing among older adults by fostering participation. The age-friendly environments network now encompasses 51 countries covering 1445 cities and communities worldwide [33]. In Norway, two cities and one municipality have committed to develop age-friendly environments and the government plan to develop more age-friendly societies in the coming years [34].

Characteristics of the built environment may play a critical role in determining health trajectories by influencing health behaviors and participation [32], yet research is currently lacking among older adults with disabilities [19]. Therefore, to deliver policies and actions designed to protect the health and well-being of this population group, there is a pressing need for current and reliable data about the relationship between the environment and associated health, well-being, and disability among older adults. This information is fundamental to develop evidence-based strategies and strengthen community action. Furthermore, the WHO has outlined the identification and measurement of environmental factors, such as accessibility to transport, nature, and services and associated secondary conditions as priority areas in disability research [35]. From Norway there are to date no known studies concerning disability and accessibility to build environment characteristics with well-being among older adults. This study therefore aims to address these knowledge gaps firstly by investigating the relationship between disability and built environment accessibility as independent predictors on older adults’ psychosocial well-being and lastly examining whether built environment accessibility plays a moderating role on psychosocial well-being among those with varying levels of disability.

## 2. Materials and Methods

### 2.1. Data

The Norwegian Counties Public Health Survey (NCPHS) is a series of online cross-sectional surveys gathering information on a broad range of health-related topics, such as physical and mental health, aspects of the lived environment, home, neighborhood and community characteristics, health behaviors and personal factors among adults aged ≥18 years old. The surveys are administered by the Norwegian Institute of Public Health (NIPH) and collect data from regionally representative samples of individuals per Norwegian county. The current investigation uses data collected 1–14 February 2021 from the NCPHS administered in Møre and Romsdal county in Western Norway. Møre and Romsdal is a region consisting of several small cities, towns, and rural villages, totaling approximately 265,000 inhabitants. Data collection took place during the third wave of the COVID-19 pandemic in Norway. Large parts of the population had taken the first vaccine dose, but social distancing was still encouraged [36].

A final sample of 54,465 study participants were drawn from the National Population Register, after screening for deceased, missing contact information, and those registered with a primary address outside of the included region. Contact information was obtained from the Norwegian Digital Agency, and invitations to voluntarily complete the online survey were sent by email and SMS. To access the survey, respondents provided proof of identity using their electronic identification details. The survey was completed by 24,967 participants (response rate = 45.8%). The data used in the present investigation comprise a sub-sample (*N* = 8274), consisting of all survey respondents aged ≥60 years. This inclusion criterion was decided on the basis that the onset of older age is generally demarked at 60 years old and in accordance with international consensus [6].

### 2.2. Variables

#### 2.2.1. Independent Variables

*Disability* was comprised from a composite of four questions. Participants were first asked if they have: 1(a) any permanent, seasonal or occasional long-term diseases/health problems lasting >6 months (yes/no) and 2(a) a functional impairment or problems due to injury (yes/no). If participants answered “yes” to any one of the two questions, for each question, they were immediately prompted to rate how the respective problem affected their daily life. Response categories were given on a 4-point scale (1 = large degree, 2 = some degree, 3 = small degree, and 4 = not at all). The disability composite variable was thus composed based on participants’ answers representing the subjective experience of reduced daily life participation in relation to a chronic health problem, and in line with WHO consensus [37]. Levels of disability were applied to recode answers using the following procedure. Firstly, answers “no” to 1(a) and 2(a) were merged with “not at all” in 1(b) and 2(b) respectively. Secondly, answer categories “small degree” and “not at all” were merged into a single category. Thirdly, levels of disability were recoded into three groups: “severe disability”, “moderate disability”, and “mild or no disability”. We decided to merge “small degree” and “not at all” into a single level based on guidelines that a small degree of reduced participation in daily life can be expected among populations [37]. Scale values were applied to the answers, and we merged 1(a)(b) and 2(a)(b). If a participant answered “large degree” on at least one of the two questions, they were assigned to scale 1: severe disability. If they reported “some degree” on at least one of the two questions, and not “large degree” on any of them, they were assigned to scale 2: moderate disability. If they only reported “small degree/not at all” on the two questions, they were assigned to scale 3: mild or no disability. A flow-diagram of the survey questions and description of the algorithm used to recode the composite disability variable is presented as Appendix A.

*Built environment accessibility* was calculated as a composite variable using the mean scores from seven questions. Participants were asked if they found specific characteristics from the built environment easily accessible/well developed. The seven questions measured accessibility to: *services* (shops, restaurants, cinemas, library, culture houses, concerts, theatres, sports halls, swimming pools, fitness centers, dance classes, ski trails, and other services), *nature* (nature areas, outdoor areas, parks and other green areas, coastlines, beaches, and the sea), and *transport* (public transport and pedestrian and bicycle paths). Responses for all items were reverse-coded on a 5-point scale (1 = very poor; 5 = very good) where higher score indicates better accessibility.

#### 2.2.2. Dependent Variables

Psychosocial well-being can be defined as the positive experience and evaluation of one’s life and social interactions [38]. *Quality of life* was measured using the combined mean score from two questions that captures cognitive and eudaimonic dimensions of life quality [39]. Participants were asked to rate on a scale from 0 = low–10 = high: “How satisfied are you with life at the moment?” and “To what extent do you find that what you do in life is meaningful?”. Higher scores indicate greater quality of life. *Thriving* was measured using the mean score from a single question, “To what extent do you thrive in your local community?”, with response categories given on a 4-point scale from 1 = low–4 = high. *Loneliness* was measured using the mean score of the validated Three-Item Loneliness Scale (T-ILS) [40]. Participants were asked three questions: “How often do you feel…(i) that you lack companionship?: (ii) left out?: (iii) isolated from others?”. Response categories were given on a 5-point scale (recoded 1 = low; 5 = high), where higher scores reflect higher levels of loneliness. Finally, *psychological distress* was measured using the combined mean score of the validated five-item Hopkins Symptom Checklist (HSCL-5) [41]. Participants were asked five questions: “In the last week, to what extent have you felt…(i) nervousness or shakiness inside?: (ii) fearful?: (iii) hopeless about the future?: (iv) blue?: (v) you are worrying too much about things?”. All questions gave response categories measured on a 4-point scale (recoded 1 = low; 4 = high), where higher scores indicate higher levels of psychological distress. Cronbach’s alpha was estimated for all multiple-item variables for the current sample, with scores 0.83 for quality of life, 0.84 for loneliness, and 0.88 for psychological distress suggesting good levels of internal consistency for all three scales.

#### 2.2.3. Demographic Variables

Participants’ *sex* and *age* were obtained from the national population registry. Age was recoded into five categories based on 5-year increments with participants aged ≥80 included in one category. Participants were asked their *partner status* and response categories were recoded into three groups: “married or cohabitating”, “non-resident partner”, and “single”. *Financial situation* as an indication of socioeconomic status was assessed using a single question: “How easy or difficult is it for you to make ends meet on a daily basis with your household income?”. Answer categories were recoded into three categories: “difficult”, “quite easy”, and “easy”. These are characteristics shown to influence psychosocial well-being and are therefore adjusted for as covariates in the present study.

#### 2.2.4. Statistical Analysis

All analyses were performed using IBM SPSS Statistics Version 27.0 for Windows (IBM Corp., Armonk, NY, USA). Descriptive characteristics were calculated for the total sample and separately for the three levels of disability. Categorical variables were reported as frequencies (n) and proportions (%), whilst continuous variables were reported as means (M) and standard deviations (SD). The Pearson chi-square test and one way analysis of variance (ANOVA) were carried out to explore statistical significance between proportions and groups of categorical and continuous variables, respectively. General linear modelling (UNIANOVA) was performed to examine the impact of the independent variables (built environment accessibility and disability) and their interaction on the dependent variables. Crude models and models adjusting for age, sex, partner status, and financial situation were performed. The Levene’s test of equality of error variance was run to test one of the assumptions underlying the ANOVA. Indication of statistical significance was set at alpha level 0.05, unless the Levene´s test was significant, in which case a more stringent significance level at 0.01 was applied. Effect sizes are reported as partial eta squared (η^2^). As a benchmark for interpreting the partial eta squared effects size, 0.01 = small effect, 0.06 = medium effect, and 0.14 = large effect [42].

## 3. Results

### 3.1. Participants’ Characteristics

Just over half of the sample participants were men (*n* = 4417, 53.4%). The mean age of the sample was 68.6 years old (SD = 6.2) and the median age was 68 years old (minimum = 60; maximum = 97). The overall response rate across all age groups (≥18 years) in this survey was 45.8%. The response rate was highest in the age group 60–69 years (59.0% women and 56.5% men), and it was lower in the age group 70+ (45.6% women and 54.0% men) [43]. The majority were married or cohabitating and reported easy financial situation.

### 3.2. Descriptive Statistics

Table 1 shows prevalence of chronic health problems, reduced participation, and disability among the sample, stratified by sex. More women reported having a chronic disease or health problem than men. Findings for the disability composite variable reveal a mean score of 2.5 (SD = 0.7). Prevalence of “severe” and “moderate” disability was highest among women.

Table 2 shows the descriptive statistics stratified by levels of disability. Higher levels of disability were most prevalent among those in the oldest and youngest age categories. Of all the age-categories, proportions of “severe” and “moderate” disability were highest among those aged ≥80 years and lowest among those aged 70–74 years. Among the other covariates, prevalence of disability was highest among participants that were single and those that reported “difficult” financial status. *Built environment accessibility* revealed that a quarter of the sample (25%) reported “very good” accessibility, and similar proportions were observed among those that answered “good” (18.6%), “neither good nor poor” (19.7%), and “poor” (20%). A total of 16.8% of the sample reported “very poor” built environment accessibility. A significant relationship was observed between scores for built environment accessibility and all levels of disability. Lower mean scores of built environment accessibility were observed among those reporting more disability, compared to less disability. Significant relationships were observed between all *psychosocial well-being* variables and all levels of disability. Less favorable scores on quality of life, thriving, loneliness, and psychological distress were observed for those with more disability, compared to those with less disability.

### 3.3. Interaction Effect between Disability and the Built Environment on Indicators of Well-Being

Table 3 shows the main and interaction effects for the outcome variables. No substantial differences were found in the crude models compared to the models adjusting for age, sex, partner status and financial situation. The Levene´s test of equality of variance on quality of life, thriving, loneliness, and psychological distress was statistically significant, suggesting the variance of these variables across the groups is not equal. A more stringent significance level at 0.01 is therefore adopted when evaluating these results.

Table 4 shows the mean values and 95% CI for estimated marginal means of outcome variables, adjusting for levels of disability and built environment accessibility.

Figure 1 shows the pattern of estimated marginal means (EMMs) adjusted models for the dependent variables across levels of disability as a function of built environment accessibility. The interaction effect between disability and built environment accessibility on *quality of life* (a) was not statistically significant (*F*(8, 5956) = 1.03, *p* = 0.414, partial eta squared (η^2^) = 0.001). However, the main effects of both disability (*F*(2, 5956) = 270.19, *p* < 0.001, η^2^ = 0.073) and built environment accessibility *(F*(4, 5956) = 54.74, *p* < 0.001, η^2^ = 0.031) were statistically significant. The interaction effect between disability and built environment accessibility on *thriving* (b) was statistically significant (*F*(8, 5936) = 4.97, *p* < 0.001, η^2^=0.006). Additionally, statistically significant main effects were observed for both disability (*F*(2, 5936) = 22.57, *p* < 0.001, η^2^=0.007) and built environment accessibility (*F*(4, 5936) = 137.89, *p* < 0.001, η^2^ = 0.075). The interaction effect between disability and built environment accessibility on *loneliness* (c) was not statistically significant after applying a more stringent alpha level (*F*(8, 5951) = 2.04, *p* = 0.038, η^2^ = 0.002). However, findings were statistically significant for the main effect for disability (*F*(2, 5951)=116.35, *p* < 0.001, η^2^ = 0.033) and built environment accessibility *(F*(4, 5951) = 56.73, *p* < 0.001, η^2^ = 0.032). In the final analysis, the interaction between disability and built environment accessibility on *psychological distress* (d) was statistically significant (*F*(8, 5957) = 3.09, *p* = 0.002, η^2^ = 0.004). In addition, statistically significant main effects were observed for both disability (*F*(2, 5957) = 202.85, *p* < 0.001, η^2^ = 0.056) and built environment accessibility (*F*(4, 5957) = 16.91, *p* < 0.001, η^2^ = 0.010).

## 4. Discussion

The purpose of this study was to investigate the relationship between disability and built environment accessibility as independent predictors on older adults’ psychosocial well-being. This study also aimed to examine whether built environment accessibility plays a moderating role on psychosocial well-being among those with varying levels of disability. Three major findings were observed. Firstly, higher levels disability were consistently associated with more unfavorable scores for quality of life, thriving, loneliness, and psychological distress. Secondly, higher mean scores for built environment accessibility were consistently related to higher psychosocial well-being and less disability. Thirdly, built environment accessibility significantly moderated the relationship between disability and two indicators of psychosocial well-being: thriving and psychological distress. These findings held true both before and after adjusting for potential confounders.

A linear relationship is apparent between higher disability and increasingly unfavorable levels of quality of life, thriving, loneliness, and psychological distress. These findings are consistent with the literature and demonstrate that presence of secondary health conditions are common among those with disabilities [10,11,12,13,14]. Several factors might explain this pattern. The experience of disability is the product of multiple factors, including personal characteristics and the extent the environment fosters positive health behaviors, social connections, independence and meaningful participation. Therefore, it is plausible that environmental factors such as services, transport, and nature that are shown to support healthy ageing [20,21,23,25,28,29,30] could partly explain this association. Higher levels of disability might influence personal needs, such as increased dependence on public transportation to attend appointments, accessible services to maintain independence, and well-equipped outdoor areas to maintain mobility. Possible underlying reasons for higher disability therefore might be due to these needs going largely unmet and highlights the importance of considering diverse needs in environmental planning.

A pattern is also observed between built environment accessibility as a predictor to psychosocial well-being and disability. Descriptive analyses displayed consistent associations between better built environment accessibility and lower levels of disability, whilst poorer environment accessibility was associated with lower psychosocial well-being and higher disability. These findings agree with prior studies observing associations between characteristics within age-friendly environments and mental health and participation [20,21,23,25,28,29,30]. Evidence indicates that older adults tend to use services and facilities more if they are easily accessible both in terms of proximity and in universal design [44]. Additionally, people tend to rate their environment as better if they are able to use it for meaningful purposes [45]. This suggests that accessible environments hinge on the interrelation between a person’s intrinsic needs and the subjective experience of how well the environment meets those needs. 

Interestingly, we found that better built environment accessibility reduced the relationship between disability and two indicators of psychosocial well-being: thriving and psychological distress. We should be quick to note here that this is a cross-sectional study that precludes inferences about causality. Additionally, the effect sizes for the interaction effects were small. However, according to the prevention paradox, a small effect of measures aimed at larger populations entails greater overall change than a large effect of measures aimed at few individuals [46]. The greatest effect of preventive measures is thus achieved by universal strategies and measures that have a broad impact. Ensuring sufficient accessibility to services, transportation, and nature for inhabitants—and perhaps especially for vulnerable groups such as older adults and those with disabilities—might be good examples of universal strategies for political and local planning work with a broad impact.

The concept of thriving communities may be understood as a community in which people can access and enjoy basic needs—for example food, clothing, shelter, and good health—as well as having opportunities for growth and success [47]. We found that better accessibility did the most to improve thriving among those who reported severe and moderate disability, compared to those who reported mild or no disability. This suggests that compared to those with less disability, those with higher disability may be more dependent on their built environment to maintain basic needs, and, thus, when accessibility is perceived as poor, this negatively influences the extent to which they thrive in their local community. This interpretation is echoed in a recent Swedish study that found positive associations between higher thriving and social participation and higher thriving and feelings of self-management among home-dwelling older adults [48]. Previous studies have also shown that accessibility to community facilities is associated with greater levels of physical activity [49,50] and opportunities for voluntary work [51], which may in turn have a positive impact on aspects of thriving. Therefore, thriving may reflect a positive interaction between a person’s ability to maintain social and functional needs in their immediate environment.

Psychological distress reflects the extent to which a person experiences symptoms of depression and anxiety [41]. Much of the same explanation for the findings around thriving will also be relevant when interpreting the results on psychological distress. In particular, there seems to be a well-established link between accessibility and greater levels of physical activity [52,53]. Indeed, measures that encourage physical activity are one of the most effective ways to improve psychological health in a public health perspective [54]. Furthermore, public green spaces and leisure facilities have been shown to have a positive impact on mental health in a dose-response manner [55], including having restorative benefits [56]. The current study suggests that this also might apply to older adults with higher levels of disability.

Similar interaction effects to those described above were not observed for loneliness and quality of life. This is contrary to the general impression from the previous research referred to above. It is unclear why we did not find the same pattern of result for loneliness and quality of life; however, it may be appropriate to speculate on one possible reason. The data collection for this study took place during the third wave of the COVID-19 pandemic in Norway, when social distancing was still encouraged and the need for transportation, services, and facilities might have been limited, which may have affected some of the results in this study. More unfavorable levels of loneliness and quality of life among older adults were found to increase over time during the pandemic in Norway [57]. It is conceivable that these outcomes were affected by the special conditions during the pandemic and showed less variation in relation to accessibility.

The topics investigated in this study are important to consider and highlight the need to target interventions that promote inclusivity and health equity. Monitoring trends in population health and associated health determinants is an important part of public health and may help towards planning future resource allocation and services targeted for older populations. Better built environment accessibility may promote resiliency and thriving and protect against psychological distress among older adults, especially for those with disabilities. These analyses provide reliable and recent assessments that should be used to help policy makers and stakeholders prioritize and address challenges concerning the well-being of older adults with disabilities, as an integral component towards achieving the Sustainable Development Goals. Measures to improve accessibility to ensure universal design to services, transport, and nature could be effective in meeting diverse physical, psychological, and social needs among older adults and may contribute to improve well-being and participation. Whilst this study focused on the role of services, transport, and nature as aspects of age-friendly environments, there is still a knowledge gap in disability research exploring the role environmental factors plays on health equity and participation. Specifically, little is known concerning how contextual factors over time influence these outcomes in a life course perspective, namely among younger adult cohorts, and the role this may play on healthy ageing in later years. More longitudinal research is therefore needed to better understand the relationship between different environmental contexts and health equity among populations with disabilities. Lastly, it is important to investigate these predictors in different settings, as international and regional differences are important to consider when examining social health gradients among populations.

A major strength of this study is the large sample size from a representative sample of the older adult population. The response rate was high among the participants in this study (over 50% for many of the included age groups), which is generally high for such surveys and strengthens the representativeness of the findings. Nevertheless, the fact that around half of those invited did not participate creates some uncertainty about the representativeness. However, analysis on data from a similar survey in Norway showed that those who respond after one or two reminders did not differ significantly from those who respond at the first invitation [58]. The authors concluded that this might suggest that the selection problem may be less than one could fear in these surveys [58]. The large sample in the NCPHS enables analyses on subgroups, such as older adults as demonstrated in the present investigation, while maintaining a decent sample size and sufficient statistical power to detect small differences and study interaction effects. However, there are also pitfalls associated with this, where large samples increase the risk of finding statistically significant results that may have little practical value.

There are also important limitations that must be considered when interpreting the results. Levels of disability were measured based on the subjective experience of reduced participation in daily life (inherently influenced by environmental and personal contexts) in relation to a chronic health problem. It must be noted that the absence of controlling for an extensive list of environmental factors curtails to some extent the validity of the disability levels used in the present investigation. Additionally, the binary ”yes/no” questions used in this study (for example, “Do you have a functional impairment or problems due to injury?”) have been criticized for leading to underreporting of health problems [59]. However, this was followed up with a question that included a finer grading of how the problem affected daily life, which might improve the validity of the question. The questions that make up the disability composite variable in this study focus on impairment from disease, health problems, and injury. However, it remains unclear how well these questions succeed in capturing those with health conditions or impairments acquired at birth. Another limitation is that the cross-sectional design cannot infer cause and effect, and so it is not possible to conclude that built environment accessibility leads to improved health outcomes based on these study findings. Although a strength in some respects, the subjective nature of the data also has some limitations related to cognitive biases and the validity of the measurements. Most of the outcome variables in this study were based on standardized questionnaires or established measurement methods for the respective phenomenon [60]. These scales also had high internal consistency in this study, which is a testament to their reliability. However, thriving was measured with a single-item scale developed for the NCPH surveys. This entails several uncertainties related to the fact that we do not know how well the scale captures the construct it is meant to measure (content validity), we have few points of discrimination (sensitivity), and we lack a measure of the scale’s reliability. Whilst this study used data on transport, services, and nature from a single, and somewhat rural, Norwegian county, it is important to note that subjective accessibility in more populated areas, both regionally and internationally, might yield different results, due to differences in available resources. Cognitive biases might also to some extent affect the measurement of built environment accessibility (such as availability heuristic and recall bias). A formative measurement model is assumed for the composite score of built environment accessibility, as opposed to a reflective model, where we do not expect the existence of an underlying construct that drives the indicators to provide high intercorrelations. Instead, people’s reported accessibility is likely to reflect the actual context/environment in which they live, which cannot be expected to follow a preconceived pattern of intercorrelation.

## 5. Conclusions

In this study, older adults with higher levels of disability tended to report lower psychosocial well-being, and this was most evident among those also reporting poorer built environment accessibility. These findings suggest that good accessibility to services, transportation, and nature is associated with better thriving and lower levels of psychological distress among older adults and those experiencing more disability. We argue that improved health outcomes are partly reliant on improving built environments at the policy level, aimed towards meeting the diverse needs of the citizens in which they serve, through ensuring universal design and age-friendliness. Whilst this study design precludes causality, our findings support the notion that accessible built environments play an influential role on improving health outcomes and leveling the social health gradient. Furthermore, this study extends on this knowledge by demonstrating this notion among older adults with disabilities in Norway. Policy makers are encouraged to consider the importance of external factors, such as accessibility to services, nature, and transportation, when planning built environments aimed to foster healthy ageing among those with disabilities. Future research should address health gradients connected to the environment among younger adults with disabilities, to better understand how contextual factors influence health and well-being over time, and to help develop policies and actions aimed towards supporting healthy ageing throughout the life-course.

## Figures and Tables

**Figure 1 ijerph-20-05898-f001:**
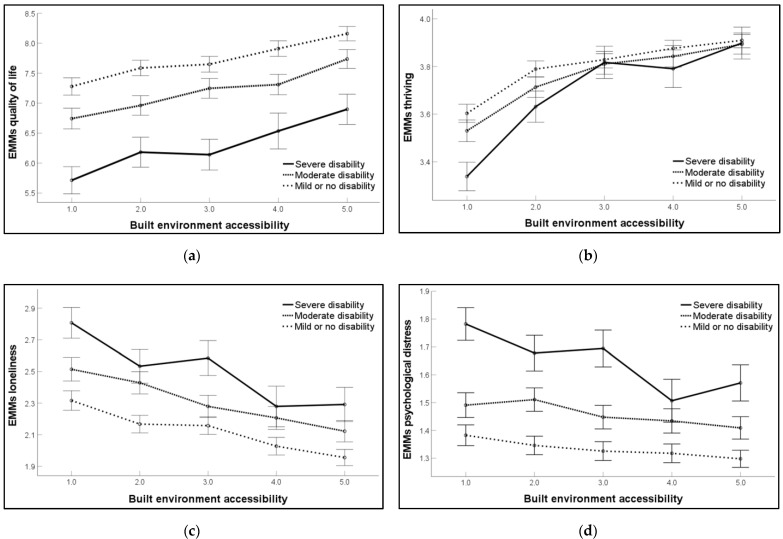
Pattern of estimated marginal means (EMMs) (adjusted models) for the dependent variables across levels of disability as a function of built environment accessibility. (**a**) Quality of life (0—low: 10—high); (**b**) Thriving (1—low: 4—high); (**c**) Loneliness (1—low: 5—high); (**d**) Psychological distress (1—low: 4—high).

**Table 1 ijerph-20-05898-t001:** Prevalence of chronic health problems, reduced participation and disability.

	Men		Women		Total		
	*n*	(%)	*n*	(%)	*n*	(%)	*p*
1(a) Chronic disease / health problems							
No	2179	(49.6)	1722	(44.9)	3901	(47.5)	<0.001
Yes	2211	(50.4)	2109	(55.1)	4320	(52.5)	
1(b) Reduced participation							
Large degree	425	(9.7)	438	(11.5)	863	(10.5)	<0.001
Some degree	1010	(23.1)	1143	(29.9)	2153	(26.3)	
Small degree or not at all ^1^	2944	(67.2)	2239	(58.6)	5183	(63.2)	
2(a) Functional impairment/problems from injury							
No	3041	(69.3)	2662	(69.3)	5703	(69.3)	0.986
Yes	1348	(30.7)	1179	(30.7)	2527	(30.7)	
2(b) Reduced participation							
Large degree	286	(6.5)	202	(5.3)	488	(5.9)	<0.001
Some degree	709	(16.2)	751	(19.6)	1460	(17.8)	
Small degree or not at all ^1^	3389	(77.3)	2887	(75.2)	6276	(76.3)	
Disability ^2^							
Severe	474	(10.7)	466	(12.1)	940	(11.4)	<0.001
Moderate	1199	(27.2)	1292	(33.6)	2491	(30.2)	
Mild or no disability	2731	(62.0)	2092	(54.3)	4823	(58.4)	

^1^ Includes participants reporting “no” to respective part (a) question. ^2^ Combined score from questions 1(a), 1(b), 2(a), and 2(b). *n* = numbers. % = proportions within total sample. Note that the composite disability variable was used for all further analyses in this paper. *p* = statistical significance, indicated with *p* < 0.05.

**Table 2 ijerph-20-05898-t002:** Descriptive statistics stratified by levels of disability.

	Severe Disability		Moderate Disability		Mild or No Disability		Total		*p*
Sex, n (%)									
Men	474	(10.8)	1199	(27.2)	2731	(62.0)	4404	(53.4)	<0.001
Women	466	(12.1)	1292	(33.6)	2092	(54.3)	3850	(46.6)	
Age groups, n (%)									
60–64	308	(12.4)	773	(31.1)	1407	(56.6)	2488	(30.1)	<0.001
65–69	286	(11.7)	726	(29.7)	1436	(58.7)	2448	(29.7)	
70–74	182	(9.8)	527	(28.4)	1147	(61.8)	1856	(22.5)	
75–79	101	(10.1)	299	(29.8)	604	(60.2)	1004	(12.2)	
≥80	62	(13.6)	166	(36.3)	229	(50.1)	457	(5.5)	
Partner status, n (%)									
Married/cohabiting	683	(10.7)	1884	(29.5)	3826	(59.8)	6393	(77.7)	<0.001
Non-resident partner	22	(6.9)	93	(29.1)	205	(64.1)	320	(3.9)	
Single	229	(15.2)	506	(33.5)	775	(51.3)	1510	(18.4)	
Financial situation, n (%)									
Difficult	216	(28.5)	283	(37.3)	260	(34.3)	759	(9.7)	<0.001
Quite easy	287	(13.0)	758	(34.3)	1167	(52.8)	2212	(28.2)	
Easy	393	(8.0)	1320	(27.0)	3172	(64.9)	4885	(62.2)	
BE accessibility (1–5), M (SD)	2.83	(1.46)	3.07	(1.42)	3.25	(1.41)	3.15	(1.43)	<0.001
Quality of life (0–10), M (SD)	6.43	(2.28)	7.58	(1.69)	8.21	(1.48)	7.81	(1.75)	<0.001
Thriving (1–4), M (SD)	3.68	(0.59)	3.79	(0.45)	3.85	(0.39)	3.81	(0.44)	<0.001
Loneliness (1–5), M (SD)	2.38	(0.92)	2.12	(0.75)	1.86	(0.67)	2.00	(0.75)	<0.001
Psychological distress (1–4), M (SD)	1.63	(0.62)	1.39	(0.45)	1.23	(0.35)	1.33	(0.44)	<0.001

BE = built environment. M = mean. SD = standard deviation. *p* = statistical significance < 0.05.

**Table 3 ijerph-20-05898-t003:** Main and interactions effects for the outcome variables.

	Crude Model			Fully Adjusted Model ^1^		
	*F*	*p*	η^2^	*F*	*p*	η^2^
Quality of life						
Disability ^2^	367.955	**<0.001**	0.093	270.187	**<0.001**	0.073
Accessibility ^2^	66.206	**<0.001**	0.035	54.737	**<0.001**	0.031
Disability * BE Accessibility ^3^	1.509	0.148	0.002	1.025	0.414	0.001
Thriving						
Disability ^2^	30.645	**<0.001**	0.008	22.565	**<0.001**	0.007
Accessibility ^2^	151.524	**<0.001**	0.078	137.891	**<0.001**	0.075
Disability * BE Accessibility ^3^	3.787	**<0.001**	0.004	4.973	**<0.001**	0.006
Loneliness						
Disability ^2^	187.785	**<0.001**	0.050	116.352	**<0.001**	0.033
Accessibility ^2^	55.212	**<0.001**	0.030	56.732	**<0.001**	0.032
Disability * BE Accessibility ^3^	1.652	0.105	0.002	2.040	0.038	0.002
Psychological distress						
Disability ^2^	307.912	**<0.001**	0.079	202.846	**<0.001**	0.056
Accessibility ^2^	23.709	**<0.001**	0.013	16.906	**<0.001**	0.010
Disability * BE Accessibility ^3^	2.940	**0.003**	0.003	3.094	**0.002**	0.004

^1^ Adjusted for age, sex, partner status, and financial situation. ^2^ Main effects. ^3^ Interaction effects. *F* = F-value. *p* = statistical significance <0.01 (highlighted in bold). η^2^ = partial eta squared.

**Table 4 ijerph-20-05898-t004:** Estimated marginal means of outcome variables, adjusting for levels of disability and built environment accessibility (adjusted model).

Built Environment Accessibility	Disability	Quality of Life (0–10)			Thriving (1–4)			Loneliness (1–5)			Psychological Distress (1–4)		
		M	SE	95% CI	M	SE	95% CI	M	SE	95% CI	M	SE	95% CI
1 = Very poor	Severe	5.71	0.12	5.49–5.94	3.34	0.03	3.28–3.40	2.79	0.05	2.70–2.89	1.78	0.03	1.72–1.84
	Moderate	6.74	0.09	6.57–6.91	3.53	0.02	3.45–3.58	2.49	0.04	2.42–2.57	1.50	0.02	1.45–1.54
	Mild or no disability	7.28	0.07	7.13–7.42	3.60	0.02	3.57–3.64	2.30	0.03	2.24–2.36	1.38	0.02	1.35–1.42
2 = Poor	Severe	6.18	0.13	5.93–6.43	3.63	0.03	3.57–3.70	2.51	0.06	2.41–2.62	1.70	0.03	1.61–1.74
	Moderate	6.96	0.08	6.80–7.13	3.71	0.02	3.67–3.76	2.41	0.04	2.34–2.48	1.51	0.02	1.47–1.55
	Mild or no disability	7.59	0.07	7.46–7.72	3.79	0.02	3.76–3.82	2.15	0.03	2.10–2.20	1.35	0.02	1.31–1.38
3 = Neither good nor poor	Severe	6.14	0.13	5.88–6.40	3.82	0.04	3.75–3.89	2.57	0.06	2.45–2.68	1.70	0.03	1.63–1.76
	Moderate	7.25	0.09	7.08–7.41	3.81	0.02	3.77–3.85	2.26	0.04	2.19–2.33	1.45	0.02	1.41–1.49
	Mild or no disability	7.65	0.07	7.52–7.78	3.83	0.02	3.79–3.86	2.14	0.03	2.08–2.19	1.33	0.02	1.29–1.36
4 = Good	Severe	6.54	0.15	6.24–6.84	3.79	0.04	3.71–3.87	2.26	0.07	2.13–2.39	1.51	0.04	1.43–1.58
	Moderate	7.31	0.09	7.14–7.48	3.84	0.02	3.80–3.89	2.19	0.04	2.11–2.26	1.43	0.02	1.39–1.48
	Mild or no disability	7.91	0.07	7.78–8.04	3.88	0.02	3.84–3.81	2.01	0.03	2.00–2.06	1.32	0.02	1.28–1.35
5 = Very good	Severe	6.90	0.13	6.65–7.15	3.90	0.03	3.83–3.97	2.27	0.06	2.20–2.38	1.57	0.03	1.51–1.64
	Moderate	7.74	0.08	7.58–7.89	3.89	0.02	3.85–3.94	2.10	0.03	2.04–2.17	1.41	0.02	1.37–1.45
	Mild or no disability	8.16	0.06	8.04–8.28	3.91	0.02	3.88–3.94	1.94	0.03	1.88–2.00	1.30	0.02	1.28–1.33

SE = Standard error. 95%CI = confidence interval of estimates (lower–upper).

## Data Availability

The dataset supporting the conclusions of this article is available only under applied licensing from the Norwegian Institute of Public Health and with ethical pre-approval by the National Committees for Research Ethics in Norway (REK). Information on the data application process and guidelines is available from https://www.fhi.no/en/more/access-to-data/applying-for-access-to-data/.

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
