# Peer review of "Built Environment Accessibility and Disability as Predictors of Well-Being among Older Adults: A Norwegian Cross-Sectional Study"

_ijerph, 2023, doi:10.3390/ijerph20105898_

Round 1

Reviewer 1 Report

This is a very well-conceived, executed, and written study. It is very rare that I have nothing to offer to improve the manuscript, but this is one of those times. The manuscript should be accepted as is and published.

Some specific comments

1. What is the main question addressed by the research?

What is the role of built environment on the well-being of individuals with disabilities?

2. Do you consider the topic original or relevant in the field? Does it address a specific gap in the field?

Relevant and important. Not necessarily original but needed. Adds to the field.

3. What does it add to the subject area compared with other published material?

Adds specific information for Norway and provides an example for others.

4. What specific improvements should the authors consider regarding the methodology? What further controls should be considered?

Some might suggest there is a lack of controls but, as is the case in most surveys, there are no controls. The survey in a national Public Health Survey with adequate respondents to ensure a diverse mixture of conditions. Age range was broad and there did not seem to be any targeting to individuals with disabilities other than it is an older cohort. Half reported no disease conditions and 60% reported no or mild disabilities.

5. Are the conclusions consistent with the evidence and arguments presented and do they address the main question posed? Yes

6. Are the references appropriate? Seems reasonable.

7. Please include any additional comments on the tables and figures. A lot of them and not always well explained but I could figure them out.

Reviewer 2 Report

The main question addressed by the research is ‘whether and how the built environment plays a role on well-being among older adults with disabilities’. The study intends to fill a knowledge gap about the influence of these factors ‘to deliver policies supporting healthy ageing and sustainable health equity’. By themselves, these aspects reveal the relevance of the article, even more so in view of the assertion that ‘from Norway there are to date, no known studies concerning disability, well-being and the built environment among older adults’. But its level of originality is reduced by the previous predictability of the outcomes. Based on official data collected in February 2021, the research design is appropriate, but an item to consider is its development during the occurrence of the third wave of the coronavirus disease 2019 (Covid-19) without deepening the critical analysis of the likely interferences of the restrictions imposed on society during the pandemic period, even more in view of the exam of the ‘interaction between built environment accessibility (services, transportation, nature) and disability, on psychosocial well-being (quality of life, thriving, loneliness and psychological distress)’. Also the introduction can be improved with more information about general characteristics of built environments related to the questions asked. This is one of the main improvements should be considered regarding the methodology, including as a control. These aspects are not clarified, making it difficult to interpret the exposed results, generally obvious, with rare new scientific findings, which is confirmed by the citations of other published materials themselves, that is, with few additions to the subject area. On the other hand, despite the content of the conclusion being consistent with evidence and arguments presented and addressing the main question posed, the highlights of several limitations of the work, with indicatives for future studies, could be more detailed in this conclusive section, expanding contributions to the knowledge field. Even with the large amount of institutional references, they are generally appropriate. A proportion greater than 60% corresponds to the last quinquennium, with around 30% up to a decade and close to 10% over 10 years. If possible, these proportions could further favor the character of actuality of the sources. English language and style are good, but a reduced spell checking is recommended, including the elimination of excessive repetitiveness of contents in general and of similar terms in the same sentences and paragraphs. Finally, it is suggested to check the arrangement of figures and tables in relation to the text because some precede their textual reading. Occasionally more graphs would make some table data easier to interpret. 

Reviewer 3 Report

Interesting article based on a large database. But I have some (mostly methodologically) comments:

The introduction is far too long, please make it more concise and more focussed on addressing the knowledge gap that this paper is going to address. Please also adress the dependent variables you use. They are now first described in the methods

the part regarding disability (line 148-168) is confusing. A flow-diagram of the algoritme you use could provide a graphical overview of how categories are formed. 

Please restructure the part of the dependent variabels. They are not introduced in the introduction of the manuscript so it is unclear why you choose these as main outcome measures. In addition, the scales you use they seem to be not validated. Measureing cronbach alphas on scales consisting of two items is not that informative. 

Please describe how the variable built environment was placed in the model. Did you use it as a categorical variable or a continues variable? 

Did you check the assumptions for the ANOVA and UNIANOVA? Based on the scales you used, it is expected that these variables do not follow a normal distribution.

The data you use is typically clustered data with clustering between citys and neigbourhood. Multilevel analyses are therefore advised to adjust herefore.

General: you use data from relative small towns for these analyses. It is likely that environmental problems will be different in larger citys as they have for instant better public transport. Please discuss this in the discussion. 

Reviewer 4 Report

The paper shows clear research and from statistical point of view is relevant and clear conclusions  

1. What is the main question addressed by the research?

The relationship between the environment with the well-being and the adults with disabilities 

2. Do you consider the topic original or relevant in the field? Does it
address a specific gap in the field?

From my point is original since the paper studies with the relationship in adults with special characteristics,  

On the other hand, from statistical point of view is relevant due to N=8274 

3. What does it add to the subject area compared with other published
material?

As I said before the special study about relationship well-being/environment and disabilities

4. What specific improvements should the authors consider regarding the
methodology? What further controls should be considered?

Probably including economic variables to check the relationship with well-being.

5. Are the conclusions consistent with the evidence and arguments presented
and do they address the main question posed?

Yes more appropriate. 

6. Are the references appropriate? 

Good references and more accurate 

7. Please include any additional comments on the tables and figures.

Reviewer 5 Report

Thank you for opportunity for reviewing this paper. The topic of the manuscript is indeed important area for good quality studies. The research area is indeed very interesting for a potential reader.

I believe that this manuscript doesn´t qualify for acceptance at this time and should be improved for publication. Than, it can be considered for acceptance.

Specific comments:

1.      Writing

The English used is appropriate.

 2.      Title

The title reflects the content and problem studied. The title reflects what type of study is presented.

Nonetheless, the title reflects “The built environment(…)”, while in the text of the manuscript it is specified that the role of the “built environment accessibility” was examined. Therefore, I suggest updating the title.

3.      Abstract

The abstract reflects the manuscript and provide a summary of what was done and what was found.

Geographical location of the respondents should be mentioned as the research covered not whole Norway, but 1 of the 11 Norway counties.

4.    Key Words

The keywords are representative of the subject of the study. Nonetheless, the number of keywords is extensive. I suggest removing either one: “aging” or “healthy ageing”. Please also consider splitting “health” from “well-being”.

5.      Introduction, Discussion, Limitations

The parts reflect the state of the art in relation to the study. Especially Introduction is very informative for a potential reader.

I may suggest adding more than one source for the strong statement used:

·         “Differences in 42 health outcomes among the general population and people with disabilities are profound, 43 with the latter more likely to have poorer health, higher risk of mortality and experience 44 more social stigma and discrimination [1]”

 I find a limitation of the study that the research covered only 1 Norway county. The reader does not know why the county was chosen, not any other? Is the built environment accessibility same, as in other Norway counties? Is the number of peoples with disabilities same or higher or lower than in other counties?

6. Methods

I recommend adding additional information:

·        -  why only 1 county was chosen in the research, as the Norwegian Institute of Public Health “collect data from regionally representative samples of individuals per Nor-131 wegian county”?

·      -   table 1 with baseline demographics, structure of population presented

7. References

The references are used correctly.

Round 2

Reviewer 3 Report

No future comments